# Association of mean corpuscular volume with 28-day mortality in sepsis patients: A retrospective cohort study using eICU data

**Huizhen Tang**[1]⊚*, **Mingli Qu**[1]‡, **Miaomiao Xin**[2]‡, **Tongqiang He**[3]⊚

**1** Department of Transfusion, Northwest Women's and Children's Hospital, Xi'an, China, **2** Reproductive Center, Northwest Women's and Children's Hospital, Xi'an, China, **3** Obstetrics and Gynecology Intensive Care Unit, Northwest Women's and Children's Hospital, Xi'an, China

⊚ These authors contributed equally to this work.
‡ MQ and MX also contributed equally to this work.
* hexbfyfck@163.com

## Abstract

### Introduction

The issue of mortality due to sepsis remains a significant concern in the field of medicine. Previous researches have demonstrated an association between mean corpuscular volume (MCV) and mortality from a range of diseases. The objective of this study was to investigate the relationship between MCV and the risk of mortality from sepsis in a large multicentre cohort.

### Method

A retrospective cohort study was conducted using data from the eICU Collaborative Research Database from 2014–2015. MCV was determined within the initial 24 hours of ICU admission, with patients subsequently classified into quartiles based on their MCV levels. Multivariate regression models were employed to investigate the correlation between MCV and 28-day mortality, with adjustments made for potential confounding factors such as age, sex, body mass index, vital signs and comorbidities. To evaluate the dose-response relationship between MCV and 28-day mortality in patients with sepsis, smoothed curve fitting and threshold effects analysis were utilised.

### Results

A total of 9,415 patients with sepsis were included in the study and the 28-day ICU mortality rate of the sepsis patients was 9.38% (883/9415). After adjusting for confounding variables, it was found that the odds ratio (OR) for 28-day mortality was 1.11 (95% CI 1.01, 1.23, P=0.04) increased followed by each 1 fl increase in MCV. The smoothed fitted curves demonstrated a nonlinear positive correlation between MCV and 28-day mortality. The inflection point for the level of MCV was 83 fl. At MCV <83 fl, there was a significant increase in the risk of 28-day mortality with each 1 fl increase in MCV (OR 1.10, 95% CI 1.02, 1.17, *P*=0.004).

**Data availability statement:** All data in this research were obtained from the publicly available eICU Collaborative Research Database. These data are freely accessible at https://eicu-crd.mit.edu/.

**Funding:** This work was supported by the Shaanxi Province Key Research and Development Programme (2024SF-YBXM-230). Tongqiang He, the host of the fund project and the corresponding author of this article, played a crucial role in study design, formal analysis, review & editing of the manuscript, and decision to publish.

**Competing interests:** The authors have declared that no competing interests exist.

## Conclusions

There is a non-linear positive correlation between MCV and 28-day risk of death in patients with sepsis. Clinicians should be aware of changes in this indicator, especially in patients with high MCV levels.

## Introduction

Mean corpuscular volume(MCV) is a significant hematological parameter utilized to evaluate the mean size of red blood cells, typically quantified in femtoliters (fl). The normal range of MCV for adult males is 80–100 fL, while for adult females it is 75–95 fl [1]. The extanting evidenced indicates that the correlation between MCV and all-cause mortality in patients with kidney failure exhibitted geographical variation. In Chinese patients, a lower MCV (less than 94 fl) was associated with an elevated risk of mortality [2]. Conversely, in Swedish patients, a lower MCV was related to a reduced risk of mortality, particularly in the context of cardiovascular disease-related deaths [3]. This suggested that the prognostic value of MCV may be subject to geographical and population-specific influences.

Sepsis is a systemic inflammatory response syndrome that arises from an infection. It is characterised by an uncontrolled immune response that results in tissue damage and organ dysfunction [4]. Progression to severe sepsis, septic shock, and multiple organ dysfunction syndrome (MODS) is a potential outcome, with mortality rates for these conditions being high [5,6]. Globally, approximately 1,400 individuals perish daily as a result of sepsis, with an estimated mortality rate of 20% [7]. In the United States, approximately 750,000 cases of sepsis occur annually, resulting in around 215,000 deaths, with a mortality rate of 28.7% [8]. In paediatric intensive care units, sepsis represents a significant cause of mortality, constituting a considerable threat to the health of children [9]. Notwithstanding advances in medical technology and greater awareness in developed countries, which have resulted in a reduction in mortality rates, sepsis remains a leading cause of death in intensive care units, particularly among critically ill patients [10,11].

Systemic inflammatory responses and oxidative stress in patients with sepsis may affect MCV by disrupting erythrocyte membrane stability or altering bone marrow hematopoietic function [12,13]. Elevated MCV may reflect a state of chronic inflammation or nutrient deficiency that exacerbates organ dysfunction [13,14]. Nevertheless, these studies have not directly investigated the relationship between MCV and mortality risk in patients with sepsis.

In light of the aforementioned considerations, this study employed a comprehensive multicentre critical care dataset from the Philips eICU database to investigate the correlation between MCV and the 28-day mortality risk in patients with sepsis.

## Materials and methods

### Data source

Data for this study were obtained from the eICU Collaborative Research Database (eICU-CRD), a multicentre database containing more than 200,000 ICU admissions from 208 hospitals in the United States during 2014 and 2015 [15]. The database provides detailed clinical data from the eICU telemedicine programme, including demographic information, physiological readings, diagnoses (International Classification of Diseases, Ninth Edition (ICD-9) codes), and other clinical data. The data is standardised cleaned and verified by the eICU-CRD team [16].

The eICU Collaborative Research Database (eICU-CRD) comprises de-identified clinical data from the years 2014–2015, as this timeframe offers the most comprehensive and standardized dataset available at the time of analysis [17]. Subsequent updates to the database were not publicly accessible during the study design phase. The uniformity of data collection protocols across intensive care units during these years ensures minimal variability in variable definitions and measurement practices.

This study was conducted in strict adherence to the ethical standards outlined in the Declaration of Helsinki (1964) and its subsequent amendments. After completing the course "Protection of Human Research Participants" (No. 65890571), the use of the database was approved by the Institutional Review Board (IRB) of Massachusetts Institute of Technology. Because of the retrospective nature of this study and the absence of direct patient intervention, the IRB of Massachusetts Institute of Technology waived the requirement for obtaining written informed consent. The study was compliant with the safe harbor provisions of the Health Insurance Portability and Accountability Act (HIPAA) and followed the Strengthening the Reporting of Observational Studies in Epidemiology (STROBE) statement.

## Study population

This retrospective observational study included all patients with an initial diagnosis of sepsis based on International Classification of Diseases (ICD) codes from the eICU Collaborative Research Database (ICD code: A41.9) [18]. The following exclusion criteria were applied: (1) patients who had been admitted to the ICU on more than one occasion; (2) It should be considered that early death may be associated with irreversible organ failure, and that MCV, as an indicator of homeostasis, may be disturbed. Therefore, patients with ICU stays of less than 48 hours were excluded, in order to ensure sufficient observation time to assess the relationship between MCV and patient prognosis; (3) patients for whom outcome data were unavailable; (4) patients for whom data on mean corpuscular volume were unavailable within the first 24 hours of ICU admission. The study flowchart is presented in Fig 1.

## Data extraction

The data pertaining to patients diagnosed with sepsis was extracted from the eICU Collaborative Research Database (eICU-CRD) utilising PostgreSQL (version 10, www.postgresql.org). The study covariates included demographic data, vital signs, severity scores, comorbidities, blood markers, and outcomes. The following data were extracted: age, sex, body mass index (BMI), body temperature, heart rate, respiratory rate, mean arterial pressure (MAP), Acute Physiology Score III, APACHE IV score, acquired immunodeficiency syndrome (AIDS), hepatic failure, metastatic cancer, leukaemia, immunosuppression, albumin, lactate, platelets, haemoglobin, red cell distribution width (RDW), white blood cell count, ICU 28-day mortality, and hospital 28-day mortality. In the case of patients with multiple vital sign measurements or laboratory tests during their ICU stay, only the initial data from the first 24 hours after ICU admission were extracted for subsequent analysis. In the event of missing covariate values, these were represented using dummy variables.

The exposure variable was MCV, with the primary outcome being 28-day mortality in ICU.

## Statistical analysis

Continuous variables are presented as the mean ± standard deviation or as the median with interquartile range, whereas categorical variables are expressed as frequency and percentage. The normality of variables is assessed using the Shapiro-Wilk test, while homogeneity of variances is evaluated with Levene's test. In the case of normally distributed continuous variables

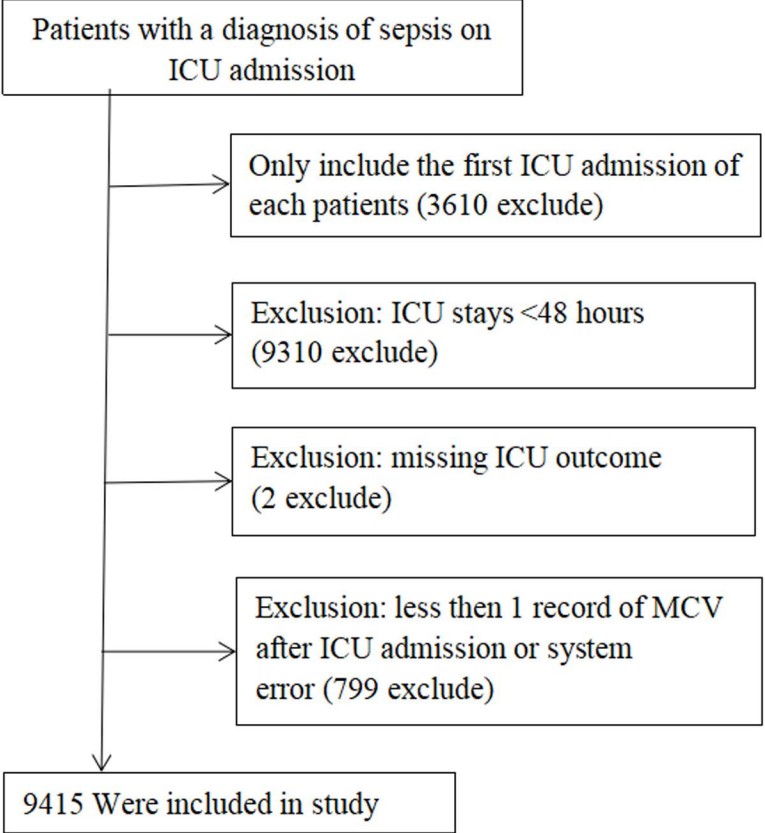

**Fig 1. Flow chart of study population.**

with equal variances, one-way analysis of variance (ANOVA) is employed; in instances of unequal variances, the Welch test is utilised. The Kruskal-Wallis H test is applied to continuous variables with a non-normal distribution. Comparisons of categorical variables are conducted using the Pearson's chi-square test or Fisher's exact test, as appropriate.

Firstly, a comparison was conducted between the various variables across the four groups (Q1–Q4). Secondly, univariate and stratified analyses were conducted to investigate the correlation between specific variables and 28-day mortality through logistic regression. For each variable, odds ratios (OR) with 95% confidence intervals (CI) were calculated. Thirdly, three distinct models were developed to explore the association between MCV and 28-day mortality. The first model was unadjusted and considered the crude association between the variables. Model I was adjusted for age and sex. Model II was fully adjusted for confounding variables. The quartiles (Q1–Q4) were employed as categorical parameters across the three models. Fourthly, the association between MCV and 28-day mortality was compared in two models: Model I (linear model) and Model II (non-linear model). The log-likelihood ratio test was employed to identify the superior model. When the p-value was less than 0.05, Model II was deemed to be significantly superior to Model I. The threshold effect of MCV in Model II was analysed, and a smoothed fitted curve was generated using a generalized additive model.

We conducted a series of sensitivity analyses to evaluate the robustness of the primary outcome. Initially, we examined follow-up outcomes in patients admitted to the ICU for less than

48 hours to determine the stability of core outcomes across varying time points. Subsequently, we adjusted for the severity of sepsis subtypes to assess their impact on the core outcomes.

Data were analyzed with the use of the statistical packages R (The R Foundation; http://www.r-project.org; version 3.6.3) and EmpowerStats (www.empowerstats.net, X&Y solutions, Inc. Boston, Massachusetts). *P* values less than 0.05 (two-sided) were considered statistically significant.

## Results

### Baseline characteristics

The entire population was divided into four groups based on the distribution of their MCV values, which were classified into quartiles. The first quartile (58.0–86.0 fl, n = 2343), the second quartile (86.1–90.9 fl, n = 2317), the third quartile (91.0–95.4 fl, n = 2384), and the fourth quartile (95.5–138.9 fl, n = 2371). The demographic characteristics of the study population across MCV quartiles revealed that age increased progressively from 62.15 ± 16.52 years in Q1 to 67.99 ± 14.56 years in Q4 (p < 0.001), while BMI decreased from 29.68 ± 10.01 in Q1 to 28.32 ± 8.49 in Q4 (p < 0.001). The proportion of males and females did not differ significantly across quartiles (p = 0.128). The 28-day mortality rates in groups Q1–Q4 were 8.41% (n = 197), 8.72% (n = 202), 8.93% (n = 213), and 11.43% (n = 271), respectively (*P* = 0.001) (Table 1).

### The results of relationship between MCV and 28-day mortality

The relationship between MCV and 28-day mortality was evaluated through the utilisation of multivariable regression models. Three models are presented in Table 2: the unadjusted model, Model I, and Model II. In Model II, which adjusted for all potential confounding variables, the odds ratio for 28-day mortality was 1.11 (95% CI 1.01, 1.23; *P* = 0.04) for each 1 fl increase in MCV. In addition, categorical variables based on MCV quartiles (Q1-Q4) were compared across the three models. In the Q4 group (95.5-138.9 fl), the greatest increase in the 28-day risk of death was observed in model II, with an OR of 1.50 (95% CI 1.10, 2.05; *P* = 0.012).

### Univariate and stratified analyses

S1 Table presented the univariate analysis of 28-day mortality in patients with severe sepsis, which showed that all factors except gender were significantly associated with 28-day mortality (*P* < 0.05). Stratified analyses showed a significant interaction between age *(P* = 0.040). The highest risk of 1.04 was found in the age group of 15–59 years (95% CI: 1.02,1.05). In addition, there was a moderating effect of body temperature (*P* = 0.050). In contrast, gender, BMI, respiratory rate, heart rate, and MAP had no significant effect on the association between MCV levels and 28-day mortality (*P*: 0.105, 0.608, 0.057, 0.121, and 0.688, respectively) (Table 3).

### Non-linear relationship between MCV and 28-day mortality

We observed a nonlinear dose–response relationship between MCV and 28-day mortality in Fig 2 (after adjusting age; gender; BMI; temperature; respiratory rate; heart rate; MAP; acute physiology score III; apache IV score; AIDS; hepatic failure; metastatic cancer; immunosuppression; albumin; lactate; platelets; hemoglobin; RDW; white blood cell count). Two distinct models were constructed for further analysis: a linear model (Model I) and a two-stage non-linear model (Model II). The results of this analysis are presented in Table 4. A significant

**Table 1. Baseline characteristics of individuals by MCV quartiles. (N=9415).**

| Characteristic | Mean CorPuscular Volume Quartile (fl) | | | | P-value |
|---|---|---|---|---|---|
| | Q1 | Q2 | Q3 | Q4 | |
| | 58.0-86.0 | 86.1-90.9 | 91.0-95.4 | 95.5-138.9 | |
| No. of participants | 2343 | 2317 | 2384 | 2371 | |
| **Demographics** | | | | | |
| Age (years) | 62.15 ± 16.52 | 65.20 ± 16.41 | 67.63 ± 15.22 | 67.99 ± 14.56 | <0.001 |
| Gender | | | | | 0.128 |
| Male | 1185 (50.60%) | 1159 (50.02%) | 1147 (48.11%) | 1131 (47.70%) | |
| Female | 1157 (49.40%) | 1158 (49.98%) | 1237 (51.89%) | 1240 (52.30%) | |
| BMI | 29.68 ± 10.01 | 29.61 ± 9.33 | 28.51 ± 8.56 | 28.32 ± 8.49 | <0.001 |
| **Vital signs** | | | | | |
| Temperature(°C) | 36.62 ± 1.33 | 36.62 ± 1.33 | 36.59 ± 1.38 | 36.51 ± 1.19 | 0.003 |
| Respiratory rate (bpm) | 31.20 ± 14.40 | 30.26 ± 14.23 | 30.81 ± 14.59 | 29.94 ± 14.76 | 0.016 |
| Heart rate (/min) | 115.26 ± 28.15 | 114.72 ± 28.29 | 114.42 ± 28.46 | 111.69 ± 29.45 | <0.001 |
| MAP (mmHg) | 56(48-112) | 56 (47-115) | 56(47-119) | 53 (45-74) | <0.001 |
| **Severity of illness** | | | | | |
| Acute Physiology Score III | 59.49 ± 25.03 | 58.87 ± 23.99 | 60.39 ± 23.91 | 62.67 ± 24.13 | <0.001 |
| APACHE IV score | 71.55 ± 26.24 | 72.20 ± 25.06 | 74.67 ± 25.01 | 77.32 ± 25.26 | <0.001 |
| **Laboratory data** | | | | | |
| Lactate(mmol/L) | 1.8 (1.2-2.93) | 1.8 (1.12-2.9) | 1.9 (1.2-3.1) | 1.9 (1.2-3.2) | <0.001 |
| MCHC(g/dL) | 32.51 ± 1.74 | 32.84 ± 1.44 | 32.68 ± 1.33 | 32.40 ± 1.53 | <0.001 |
| Platelets (x 109/L) | 205(133-291) | 195(134-273) | 182 (125-250) | 163(110-225) | <0.001 |
| Hemoglobin (g/dL) | 9.89 ± 2.01 | 10.45 ± 2.09 | 10.64 ± 2.17 | 10.38 ± 2.18 | <0.001 |
| RDW(%) | 17.06 ± 3.04 | 15.72 ± 2.19 | 15.61 ± 2.21 | 16.13 ± 2.69 | <0.001 |
| White blood cell (x 109/L) | 14.1(9.41-20.7) | 13.99 (9.3-20.1) | 13.4(8.8-19.29) | 12.9 (8.5-19) | 0.065 |
| **Comorbidities** | | | | | |
| AIDS | | | | | 0.721 |
| No | 2297 (99.65%) | 2277 (99.61%) | 2356 (99.70%) | 2338 (99.79%) | |
| Yes | 8 (0.35%) | 9 (0.39%) | 7 (0.30%) | 5 (0.21%) | |
| Hepatic failure | | | | | <0.001 |
| No | 2271 (98.52%) | 2254 (98.60%) | 2321 (98.22%) | 2256 (96.29%) | |
| Yes | 34 (1.48%) | 32 (1.40%) | 42 (1.78%) | 87 (3.71%) | |
| Leukaemia | | | | | 0.418 |
| No | 2275 (98.70%) | 2257 (98.73%) | 2335 (98.82%) | 2303 (98.29%) | |
| Yes | 30 (1.30%) | 29 (1.27%) | 28 (1.18%) | 40 (1.71%) | |
| Immunosuppression | | | | | 0.691 |
| No | 2176 (94.40%) | 2147 (93.92%) | 2237 (94.67%) | 2216 (94.58%) | |
| Yes | 129 (5.60%) | 139 (6.08%) | 126 (5.33%) | 127 (5.42%) | |
| **Outcome** | | | | | |
| ICU 28-day mortality | | | | | 0.001 |
| No | 2146 (91.59%) | 2115 (91.28%) | 2171 (91.07%) | 2100 (88.57%) | |
| Yes | 197 (8.41%) | 202 (8.72%) | 213 (8.93%) | 271 (11.43%) | |
| Hospital 28-day mortality | | | | | <0.001 |
| No | 1990 (84.93%) | 1991 (85.93%) | 2034 (85.32%) | 1921 (81.02%) | |
| Yes | 353 (15.07%) | 326 (14.07%) | 350 (14.68%) | 450 (18.98%) | |

Notes: BMI, body mass index; MAP,mean arterial pressure; RDW, red cell distribution width; MCHC, mean cellular hemoglobin concentration; AIDS, acquired immu-nodeficiency syndrome; APACHE IV acute physiology and chronic health evaluation IV; ICU intensive care unit.

**Table 2. Relationship between MCV and 28-day mortality.**

| Outcomes | Crude Model | | Model I | | Model II | |
|---|---|---|---|---|---|---|
| | OR(95%CI) | *P*-value | OR(95%CI) | *P*-value | OR(95%CI) | *P*-value |
| MCV(fl)quartile | | | | | | |
| Q1 | Reference | | Reference | | Reference | |
| Q2 | 1.04(0.85,1.28) | 0.705 | 1.00(0.81,1.23) | 0.996 | 1.40(1.01,1.96) | 0.046 |
| Q3 | 1.07(0.87,1.31) | 0.52 | 1.00(0.81,1.22) | 0.966 | 1.18(0.85,1.65) | 0.32 |
| Q4 | 1.41(1.16,1.71) | <0.001 | 1.31(1.07,1.59) | 0.007 | 1.50(1.10,2.05) | 0.012 |
| MCV(fl)quartilecontinuous | 1.12(1.05,1.19) | <0.001 | 1.09(1.02,1.16) | 0.008 | 1.11(1.01,1.23) | 0.04 |

Crude model: we did not adjust other covariants; Model I adjusted for: Age and Gender; Model II adjusted for: Age, Gender, BMI, Temperature, Respiratory rate, Heart rate, MAP, Acute Physiology Score III, APACHE IV score, AIDS, Hepatic failure, Metastatic cancer, Immunosuppression, Albumin, Lactate, Platelets, Hemoglobin, RDW and White blood cell count.

**Table 3. Effect of MCV level on 28-day mortality in stratifed analyses.**

| Characteristics | N | 28-day mortality | P for interaction |
|---|---|---|---|
| Gender | | | 0.105 |
| Male | 4622 | 1.01(1.00,1.02) | |
| Female | 4792 | 1.03(1.01,1.04) | |
| Age(years) | | | 0.040 |
| 15-59 | 2987 | 1.04(1.02,1.05) | |
| 60-73 | 3164 | 1.02(1.00,1.03) | |
| 74-89 | 3264 | 1.01(0.99,1.02) | |
| BMI | | | 0.608 |
| 10.94-24.34 | 3077 | 1.02(1.00,1.03) | |
| 24.34-30.71 | 3076 | 1.02(1.00,1.03) | |
| 30.72-55.1 | 3078 | 1.03(1.01,1.04) | |
| Temperature | | | 0.050 |
| 20-36.28 | 2689 | 1.03(1.02,1.05) | |
| 36.3-36.67 | 2642 | 1.02(1.00,1.04) | |
| 36.7-41.9 | 3511 | 1.00(0.99,1.02) | |
| Respiratoryrate(bpm) | | | 0.057 |
| 4-27 | 3000 | 1.03(1.01,1.05) | |
| 28-36 | 3088 | 1.03(1.02,1.05) | |
| 37-60 | 3168 | 1.01(0.99,1.02) | |
| Heartrate(/min) | | | 0.121 |
| 20-105 | 3008 | 1.00(0.99,1.02) | |
| 106-125 | 3123 | 1.03(1.01,1.04) | |
| 126-218 | 3158 | 1.03(1.01,1.04) | |
| MAP(mmHg) | | | 0.688 |
| 40-49 | 3066 | 1.02(1.01,1.04) | |
| 50-64 | 3082 | 1.02(1.00,1.03) | |
| 65-200 | 3132 | 1.01(0.99,1.03) | |

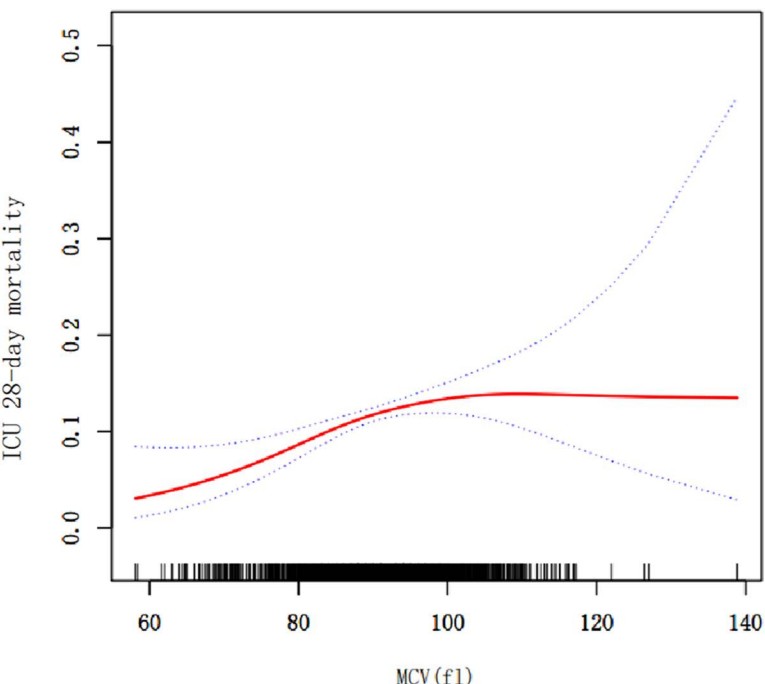

**Fig 2. Association between MCV and 28-day mortality in ICU patients with sepsis. Notes**: A threshold, nonlinear association between MCV and 28-day mortality was found in a generalized additive model (GAM). Solid rad line represents the smooth curve fit between variables. Blue bands represent the 95% of confidence interval from the fit. All adjusted for Age, Gender, BMI, Temperature, Respiratory rate, Heart rate, MAP, Acute Physiology Score III, Apache IV score, AIDS, Hepatic failure, Metastatic cancer, Immunosuppression, Albumin, Lactate, Platelets, Hemoglobin, RDW and White blood cell count.

**Table 4. Threshold effect analysis of MCV and 28-day mortality.**

| Outcome | ICU 28-day mortality | P-value |
|---|---|---|
| | OR,95%CI, | |
| Model I | | |
| Linear effect | 1.02 (1.00, 1.03) | 0.003 |
| Model II | | |
| Knot (K) | 83 | |
| Effect 1 (< K) | 1.10 (1.02, 1.17) | 0.004 |
| Effect 2 (> K) | 1.01 (0.99, 1.02) | 0.31 |
| Difference in effect (2–1) | 0.92 (0.86, 0.99) | 0.018 |
| Predicted value at knot | -2.18(-2.37, -1.99) | |
| Likelihood ratio test | | 0.011 |

Effect: ICU 28-day mortality; Cause: MCV; Adjusted: age, gender, BMI, temperature, respiratory rate, heart rate, MAP, Acute Physiology Score III, APACHE IV score, AIDS, hepatic failure, metastatic cancer, immunosuppression, albumin, lactate, platelets, hemoglobin, RDW and white blood cell count.

increase in the 28-day risk of death was observed with each 1fl increase at MCV< 83fl (OR 1.10, 95% CI 1.02, 1.17, *P*=0.004). The significant positive correlation was absent when MCV was greater than 83fl (OR 1.02, 95% CI 0.99, 1.02, *P* = 0.31).

### Sensitivity analysis

We further investigated the association between MCV and mortality at a follow-up of less than 48 hours and obtained comparable results (S2 Table). We adjusted the Severity of sepsis subtypes (septic shock and non-shock sepsis), an important factor, and found that the core results were consistent with the unadjusted results (S3 Table).

## Discussion

This study utilized the eICU multicenter critical care database to investigate the association between MCV and 28-day mortality in septic patients. The study population consisted of 9,415 patients, with data sourced from multiple ICUs, providing a robust foundation for the generalizability of the results. A major strength of the study design was the control of multiple confounding variables, including age, sex, BMI, vital signs, and severity of illness scores. The key finding revealed below the MCV threshold of 83 fl, each 1 fl increase was associated with a 10% rise in 28-day mortality risk (OR 1.10, 95% CI 1.02–1.17, P=0.004). Above 83 fl, no significant association was observed (OR 1.01, 95% CI 0.99–1.02, P=0.31), indicating a nonlinear relationship with diminishing effects at higher MCV levels.

In this retrospective cohort, sepsis was defined using the ICD code (A41.9) and included patients with systemic inflammatory response syndrome secondary to infection. The 28-day ICU mortality rate in our cohort (9.38%) was significantly lower than the global mortality rate [19]. This difference may reflect advances in earlier sepsis identification and standardized management programs, such as the Surviving Sepsis Campaign guidelines [20]. Our population consisted primarily of older adults (mean age: 65.7±15.8 years), consistent with the demographics of sepsis, where age is a recognized risk factor for adverse outcomes [21]. In addition, the inclusion of 335 intensive care unit patients from 208 hospitals in the United States enhances the generality of our findings across different critical care Settings.

The relationship between MCV and clinical disease remains a topic of contention in the scientific community. To illustrate, a 13-year Korean retrospective cohort study comprising 36,260 anaemia-free and cancer-free participants aged 40 years and older revealed that elevated MCV levels were linked to an elevated risk of all-cause mortality and cancer mortality [22]. However, an alternative study based on data from the National Health and Nutrition Examination Survey (NHANES) indicated a U-shaped relationship between MCV and mortality [23], which may be attributed to discrepancies in population characteristics and methodological approaches. Furthermore, the relationship between MCV and mortality risk is not consistent across different diseases. Hsieh et al. demonstrated that elevated MCV was significantly associated with all-cause mortality and cardiovascular mortality in patients with chronic kidney disease [24]. Similarly, a positive association was observed between MCV at admission and 30-day mortality in patients with cerebral haemorrhage using the MIMIC database [25]. Our study offers partial support for the findings of the aforementioned studies. In examining the correlation between MCV and mortality in patients with sepsis, we put forward several potential mechanisms: (1) Association between erythrocyte size and inflammation: alterations in erythrocyte morphology have been linked to inflammatory markers [26], and sepsis is a systemic inflammatory response [27]. (2) Oxidative stress and erythrocyte dysfunction: Patients with sepsis typically experience significant oxidative stress, which impairs erythrocyte deformability and endothelial function [28]. Additionally, oxidative stress disrupts haemoglobin metabolism, further exacerbating systemic hypoxia and inflammatory responses in patients [29–31].

MCV may provide independent prognostic information from the perspectives of red blood cell metabolism and chronic inflammation. Previous studies have established that elevated

lactate levels and increased SOFA scores are strong predictors of short-term mortality in critically ill patients [32,33]. In this study, we found that MCV retains significant incremental predictive value even after adjusting for lactate. While the predictive capability of MCV alone may be less robust compared to that of lactate levels and SOFA scores, its low cost and ease of accessibility render it a valuable supplementary tool for risk stratification, particularly in resource-limited environments [34]. Drawing upon the findings of this study regarding the nonlinear association, we propose the inclusion of MCV as a supplementary indicator in the current sepsis risk stratification model. Specifically, for patients exhibiting an MCV greater than 83 fL, it is imperative for clinicians to exercise heightened vigilance and undertake a thorough evaluation that integrates lactate levels, inflammatory markers, and organ function scores.

### Study advantages and limitations

The present study has certain advantages. Firstly, it was a large-sample multicentre study; secondly, we analysed MCV as both a continuous and categorical variable; thirdly, we applied a two-segmented linear model to construct a threshold-effects analysis of the relationship between MCV and 28-day mortality in sepsis patients; and we used stratified analyses to avoid, as far as possible, the occurrence of chance in the statistical analyses and to improve the stability of the results.

The current study is not without its limitations. First, a common problem in observational studies is the presence of confounding factors that cannot be measured [35]. In the current study, data on interventions in the initial stabilization phase were lacking; For example, blood transfusions may lead to elevated MCV levels and improved survival outcomes, a confounding variable that cannot be measured. Therefore, the conclusions of this study cannot be extrapolated to patients undergoing transfusion therapy.

Second, the study was conducted over a 28-day period. During this time, we excluded patients who had been admitted to the ICU for less than 48 hours, as our observations indicated that the majority of these patients were either deceased or had been abandoned. Consequently, we performed a core outcome analysis on patients admitted to the ICU for less than 48 hours, and the results were consistent with the primary findings observed over the 28-day period. However, it is important to acknowledge that these findings are based on short-term follow-up and may not be applicable to follow-up periods extending to 60 days or longer.

Finally, the database is derived from studies conducted on U.S. populations, and the absence of external validation may limit the generalizability of the findings to patient populations in other countries or regions. Future research should focus on conducting high-quality prospective studies that incorporate external validation.

### Conclusions

This multicenter retrospective cohort study involving 9,415 sepsis patients identifies a nonlinear association between mean corpuscular volume (MCV) and 28-day mortality, with a critical threshold at 83 fl. Below this threshold, each 1 fl increase in MCV is associated with a 10% increase in mortality risk, whereas no significant association is observed above this level. These findings suggest that MCV may serve as a valuable prognostic marker for risk stratification in sepsis, particularly in populations with elevated MCV. Despite the inherent limitations of retrospective analyses, this study underscores the necessity for mechanistic investigations into the role of MCV in sepsis pathophysiology and calls for validation in diverse cohorts to enhance its clinical applicability.

## Supporting information

**S1 Table. Univariate analysis for 28day mortality.**
(DOCX)

**S2 Table. Relationship between MCV and 28-day mortality during ICU stay less than 48 hours.**
(DOCX)

**S3 Table. Relationship between MCV and 28-day mortality when adjusted for sepsis subtypes.**
(DOCX)

## Acknowledgments

We would like to express our sincerest gratitude to all individuals who participated in this study, as well as to all those who contributed their efforts to this study.

## Author contributions

**Conceptualization:** Hui-Zhen Tang.

**Formal analysis:** Hui-Zhen Tang, Mingli Qu, Tongqiang He.

**Methodology:** Hui-Zhen Tang, Miaomiao Xin.

**Supervision:** Mingli Qu, Tongqiang He.

**Writing – original draft:** Hui-Zhen Tang.

**Writing – review & editing:** Hui-Zhen Tang, Tongqiang He, Miaomiao Xin.

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
