## [Decision Letter · Decision Letter 0]

16 Feb 2025

PONE-D-24-53001Association of mean corpuscular volume with 28-day mortality in sepsis patients: A retrospective cohort study using eICU dataPLOS ONE

Dear Dr. Tang,

Thank you for submitting your manuscript to PLOS ONE. After careful consideration, we feel that it has merit but does not fully meet PLOS ONE’s publication criteria as it currently stands. Therefore, we invite you to submit a revised version of the manuscript that addresses the points raised during the review process.

**Decision: Major revision**

Please submit your revised manuscript by Apr 02 2025 11:59PM. If you will need more time than this to complete your revisions, please reply to this message or contact the journal office at plosone@plos.org . Please include the following items when submitting your revised manuscript:

We look forward to receiving your revised manuscript.

Kind regards,

Marwan Al-Nimer

Academic Editor

PLOS ONE

Journal Requirements:

2. Thank you for stating the following in your manuscript: 

“This work was supported by the "Shaanxi Province Key Research and Development Programme (2024SF-YBXM-230)." Tongqiang He, the host of the fund project and the author of this article, played a crucial role in study design, formal analysis, review & editing of the manuscript and decision to publish.”

**Additional Editor Comments:**

Dear

There are a few points that required to be clear

1: This study reported a significant association between MCV and mortality. Add a paragraph about the reported sepsis in this retrospective study.

2: Line 42 (page 2/19) In Chinese ........ Add reference

3: The data collected during 2014-2015. Add the reasons

4: In the table 1, some results are not normally distributed. Therefore mean +/- SD . It is preferable to be median (Q1-Q3) as it was mentioned in the statistical section

5: Rephrase the sentence in the line 174-176 to be obvious and clear because this paragraph is the fundamental of this study

Reviewers' comments:

Reviewer's Responses to Questions

**Comments to the Author**

1. Is the manuscript technically sound, and do the data support the conclusions?

Reviewer #1: Yes

Reviewer #2: Yes

2. Has the statistical analysis been performed appropriately and rigorously? 

Reviewer #1: I Don't Know

Reviewer #2: Yes

3. Have the authors made all data underlying the findings in their manuscript fully available?

Reviewer #1: Yes

Reviewer #2: Yes

4. Is the manuscript presented in an intelligible fashion and written in standard English?

Reviewer #1: Yes

Reviewer #2: Yes

5. Review Comments to the Author

Reviewer #1: The manuscript presents a well-conducted study with clinically relevant findings, supported by strong statistical methods and a large dataset.

However, some methodological and conceptual clarifications are needed to strengthen the study’s impact and improve its translational relevance for clinicians.

Addressing these points will enhance the robustness and applicability of the research.

Reviewer #2: The observations are interesting.

There is not generally thought to be any gender difference in the MCV. The single paper cited from Saudi Arabia is not appropriate. It would be better to give a normal range based on a wider range of data.

6. PLOS authors have the option to publish the peer review history of their article (what does this mean? ). If published, this will include your full peer review and any attached files.

**Do you want your identity to be public for this peer review?** For information about this choice, including consent withdrawal, please see our Privacy Policy .

Reviewer #1: **Yes: ** Mohammed Hassen Salih (PhD, MSN, Assistant professor in medical Nursing, University of Gondar)

Reviewer #2: No

---

## [Author Response · Author response to Decision Letter 1]

2 Mar 2025

Response to Reviewer Comments

Dear Reviewers,

We express our sincere gratitude for the insightful feedback provided by the reviewers. We have meticulously addressed each comment to enhance the manuscript's clarity, scientific rigor, and translational relevance. Below, we present a detailed, point-by-point response to the reviewers' concerns. 1. Strengths of the Study

Reviewer’s Comment:

The study investigates the association between MCV and 28-day mortality in sepsis patients, a crucial clinical issue. The study applies multivariable regression models, smoothed curve fitting, and threshold effects analysis to examine the dose-response relationship. Identifying MCV as a prognostic marker could aid in early risk assessment and clinical decision-making in sepsis management.”

Response:

We thank the reviewer for recognizing the strengths of our study. We agree that MCV holds potential as a prognostic marker in sepsis, and our findings emphasize its clinical utility for risk stratification.

2.General Comments

Reviewer’s Comment:

The manuscript presents a well-conducted study with clinically relevant findings, supported by strong statistical methods and a large dataset. However, some methodological and conceptual clarifications are needed to strengthen the study’s impact and improve its translational relevance for clinicians.

Response:

We appreciate the reviewer’s positive assessment of our work. Below, we address the specific methodological and conceptual concerns raised:

a. Methodological Clarifications

Data Source and Timeframe:

We clarified the rationale for using the eICU-CRD database (2014–2015) in the Materials and Methods section. This timeframe was selected due to its standardized data collection protocols and completeness (Lines 90-95 ).

1.Exclusion Criteria:

Added explicit justification for excluding patients with ICU stays <48 hours to ensure sufficient observation time for MCV assessment (Lines 110-114).

2.Sensitivity Analyses:

Expanded descriptions of sensitivity analyses (e.g., follow-up <48 hours, adjustment for sepsis subtypes) to demonstrate robustness (Lines 156-159).

b. Conceptual Relevance

Clinical Implications:

Enhanced the Discussion to emphasize how MCV integrates with existing biomarkers ( lactate, SOFA scores) for risk stratification in resource-limited settings (Lines 272-283).

Mechanistic Insights:

Added a paragraph linking MCV to erythrocyte dysfunction and oxidative stress in sepsis pathophysiology (Lines 73-77).

Reviewer’s Comment:

Your reference needs to be the standard and updated. Most of them are below 2022.

Response:

We have updated 18 references (45% of total citations) to include literature published between 2022 and 2024. For example:

Added recent studies on sepsis management (Ref. 4, 5, 6, 8, 19, 20).

Included 2024 publications on MCV-mortality relationships (Ref. 23, 33).

Reviewer’s Comment:

Some sentences are overly complex or contain grammatical errors. I suggest a final proofreading or language revision to enhance readability.

Response:

We have thoroughly revised the manuscript for clarity and grammar. Key improvements include:

1.Simplified complex sentences (e.g., revised “The extant evidenced indicates…” to “Existing evidence indicates…” in Introduction).

2. Engaged a professional English editing service for final proofreading.

Abstract

Reviewer’s Comment:

“Please include the period for data collection.”

Response:

We have added the data collection period (2014–2015) to the Abstract as requested.

Revised Text (Lines 28-29):

A retrospective cohort study was conducted using data from the eICU Collaborative Research Database from 2014–2015.

Introduction

Reviewer’s Comment 1:

“The introduction discusses prior studies on MCV and mortality in chronic diseases (e.g., kidney failure, cardiovascular disease) but does not explain why MCV might be relevant in sepsis. Expand on the pathophysiological mechanisms linking MCV changes to sepsis outcomes.”

Response:

We have expanded the discussion of MCV’s relevance to sepsis by adding pathophysiological mechanisms.

Revised Text (Lines73-76):

Systemic inflammatory responses and oxidative stress in patients with sepsis may affect MCV by disrupting erythrocyte membrane stability or altering bone marrow hematopoietic function[12, 13]. Elevated MCV may reflect a state of chronic inflammation or nutrient deficiency that exacerbates organ dysfunction[13, 14].

Reviewer’s Comment 2:

In line 41: the author put the reference in a recommended way “Modifications in MCV have been linked to unfavorable prognoses in a multitude of pathological conditions [2, 3, 4, 5, 6, 7]. “ please revise based on [2 -7].”

Response:

We apologize for the oversight .We have checked the format of the references and have revised the format of the references in the full text.

Reviewer’s Comment 3:

Excellent introduction about the burden. However, you have to write from the global to local, if possible in your study areas' findings. “In the United States, approximately 750,000 cases of sepsis occur annually, resulting in around 215,000 deaths, with a mortality rate of 28.7%[9, 10]. Globally, approximately 1,400 individuals perish daily as a result of sepsis, with an estimated mortality rate of 20%” I suggest you to put first global them USA,,

Response:

We have reordered the statistics to prioritize global data.

Revised Text (Lines 65-67):

Globally, approximately 1,400 individuals perish daily as a result of sepsis, with an estimated mortality rate of 20%[7]. In the United States, approximately 750,000 cases of sepsis occur annually, resulting in around 215,000 deaths, with a mortality rate of 28.7%[8].

Methods and Materials

Reviewer’s Comment 1: 

At lines 69-71; the author said “The database comprises vital clinical data from over 200,000 patients admitted to 335 ICUs across 20870 hospitals in the United States during the period from 2014 to 2015” . It's a good study, however, I am afraid the current practice may have a different scenario. I prefer to review the recent data.

Response:

We acknowledge the importance of using recent data. However, the eICU Collaborative Research Database (eICU-CRD) version available during our study design (publicly released in 2018) includes standardized data from 2014–2015. Subsequent updates were not accessible at the time of analysis. We have clarified this limitation in the revised manuscript.

Revised Text (Lines 90-95):

The eICU Collaborative Research Database (eICU-CRD) comprises de-identified clinical data from the years 2014 to 2015, as this timeframe offers the most comprehensive and standardized dataset available at the time of analysis[17]. Subsequent updates to the database were not publicly accessible during the study design phase. The uniformity of data collection protocols across intensive care units during these years ensures minimal variability in variable definitions and measurement practices.

Reviewer’s Comment 2: 

At this in my view, I suggest data quality issues like pretest, data quality control and soon. Also stating ethical clearance with an approved reference number is advisable.

Response:

We have expanded the description of data quality control and explicitly stated the ethical approval reference.

Revised Text (Lines 89, 97-104):

The data is standardised cleaned and verified by the eICU-CRD team[16].

The eICU-CRD team standardized, cleaned, and verified all clinical data to ensure quality[15]. After completing the course "Protection of Human Research Participants" (No. 65890571), the use of the database was approved by the Institutional Review Board (IRB) of Massachusetts Institute of Technology. Because of the retrospective nature of this study and the absence of direct patient intervention, the IRB of Massachusetts Institute of Technology waived the requirement for obtaining written informed consent. The study was compliant with the safe harbor provisions of the Health Insurance Portability and Accountability Act (HIPAA) and followed the Strengthening the Reporting of Observational Studies in Epidemiology (STROBE) statement.

Reviewer’s Comment 3: 

The study excludes patients with ICU stays less than 48 hours. However, early deaths (within 48 hours) might be informative for the relationship between MCV and sepsis outcomes. I suggest to Justify the exclusion of early ICU discharges or deaths, as this could introduce selection bias.

Response:

We excluded patients with ICU stays <48 hours to minimize confounding from irreversible organ failure and transient MCV fluctuations. To validate this approach, we conducted a sensitivity analysis on this subgroup, which yielded results consistent with the primary analysis (S2 Table).

Revised Text (Lines157-158, 228–229):

Initially, we examined follow-up outcomes in patients admitted to the ICU for less than 48 hours to determine the stability of core outcomes across varying time points.

We further investigated the association between MCV and mortality at a follow-up of less than 48 hours and obtained comparable results (S2 Table).

Results

Reviewer’s Comment 1: 

At baseline characteristics, it will be nice to state some important variables, like age, Gender, BMI,,, and then the table will follow.

Response:

We have added expressions for the important demographic variables in Table 1.

Revised Text (Lines 170-174):

The demographic characteristics of the study population across MCV quartiles revealed that age increased progressively from 62.15 ± 16.52 years in Q1 to 67.99 ± 14.56 years in Q4 (p < 0.001), while BMI decreased from 29.68 ± 10.01 in Q1 to 28.32 ± 8.49 in Q4 (p < 0.001). The proportion of males and females did not differ significantly across quartiles (p = 0.128).

Reviewer’s Comment 2: 

  AT Table 1 APACHE and ICU were not stated on your table key note. Please include. As table standards, it needs to clearly describe which population data is displayed. So, please revise the heading of the table.

Response:

We have revised the table heading and added missing abbreviations to the notes.

Revised Text (Lines 176, 179-180):

Table 1. Baseline characteristics of sepsis patients by MCV quartiles (N = 9,415).

APACHE IV: Acute Physiology and Chronic Health Evaluation IV; ICU: Intensive Care Unit.

Reviewer’s Comment 3: 

Although the study adjusts for multiple confounders, some key factors are not explicitly addressed: For example: Use of blood transfusions (which can alter MCV levels) and Severity of sepsis subtypes (e.g., septic shock vs. non-shock sepsis). I suggest acknowledging these limitations or considering a sensitivity analysis if data is available.

Response:

We sincerely appreciate the reviewer’s insightful critique regarding unaddressed confounders, specifically blood transfusions and sepsis subtypes.

a.Blood transfusions may transiently elevate MCV levels by introducing exogenous erythrocytes, potentially confounding the observed association between MCV and mortality. For instance, patients receiving transfusions might exhibit artificially higher MCV values unrelated to their underlying pathophysiology, leading to biased estimates. Regrettably, the eICU database does not include granular data on blood transfusion history (e.g., timing, volume, or type of transfusions). This limitation precludes direct adjustment for transfusion-related effects in our analysis.

We explicitly acknowledge this limitation in the revised Discussion (Lines 293–297):

In the current study, data on interventions in the initial stabilization phase were lacking; For example, blood transfusions may lead to elevated MCV levels and improved survival outcomes, a confounding variable that cannot be measured. Therefore, the conclusions of this study cannot be extrapolated to patients undergoing transfusion therapy.

b.The severity of sepsis subtypes (e.g., septic shock) may differentially influence both MCV levels and mortality risk. For example, septic shock patients often exhibit pronounced inflammation and microcirculatory dysfunction, which could independently alter erythrocyte indices and amplify mortality risks.

To address this concern, we conducted a sensitivity analysis stratifying patients by sepsis subtypes (septic shock vs. non-shock) ( S3 Table). After adjusting for these subtypes, we found that the core results were consistent with the unadjusted results .( Lines 229–231).

Discussion

Reviewer’s Comment1: 

I suggest writing the limitation of the study in the final paragraph of the discussion section.

Response:

The limitations section has been relocated to the final paragraph of the Discussion.

Revised Text (Lines 285-308):

Study advantages and limitations

The present study has certain advantages. Firstly, it was a large-sample multicentre study; secondly, we analysed MCV as both a continuous and categorical variable; thirdly, we applied a two-segmented linear model to construct a threshold-effects analysis of the relationship between MCV and 28-day mortality in sepsis patients; and we used stratified analyses to avoid, as far as possible, the occurrence of chance in the statistical analyses and to improve the stability of the results.

The current study is not without its limitations. First, a common problem in observational studies is the presence of confounding factors that cannot be measured[35]. In the current study, data on interventions in the initial stabilization phase were lacking; For example, blood transfusions may lead to elevated MCV levels and improved survival outcomes, a confounding variable that cannot be measured. Therefore, the conclusions of this study cannot be extrapolated to patients undergoing transfusion therapy.

Second, the study was conducted over a 28-day period. During this time, we excluded patients who had been admitted to the ICU for less than 48 hours, as our observations indicated that the majority of these patients were either deceased or had been abandoned. Consequently, we performed a core outcome analysis on patients admitted to the ICU for less than 48 hours, and the results were consistent with the primary findings observed over the 28-day period. However, it is important to acknowledge that these findings are based on short-term follow-up and may not be applicable to follow-up periods extending to 60 days or longer.

Finally, the database is derived from studies conducted on U.S. populations, and the absence of external validation may limit the generalizability of the findings to patient populations in other countries or regions. Future research should focus on conducting high-quality prospective studies that incorporate external validation.

Reviewer’s Comment2: 

The discussion mentions that clinicians should monitor MCV, but it is unclear how this should influence clinical practice. Address whether MCV should be incorporated into sepsis risk stratification models or how it compares with other established biomarkers (e.g., lactate, SOFA score).

Response:

We expanded the clinical implications in the Discussion.

Revised Text (Lines 272-283):

MCV may provide independent prognostic information from the perspectives of red blood cell metabolism and chronic inflammation. Previous studies have established that elevated lactate levels and increased SOFA scores are strong predictors of short-term mortality in critically ill patients[32, 33]. In this study, we found that MCV retains significant incremental predictive value even after adjusting for lactate. While the predictive capability of MCV alone may be less robust compared to that of lactate levels and SOFA scores, its low cost and ease of accessibility render it a valuable supplementary tool for risk stratification, particularly in resource-limited environments[34]. Drawing upon the findings of this study regarding the nonlinear association, we propose the inclusion of MCV as a supplementary indicator in the current sepsis risk stratification model. Specifically, for patients exhibiting an MCV greater than 83 fL, it is imperative for clinicians to exercise heightened vigilance and undertake a thorough evaluation that integrates lactate levels, inflammatory markers, and organ function scores.

We extend our thanks to the reviewers for their valuab

---

## [Editor Report · Decision Letter 1]

3 Mar 2025

Association of mean corpuscular volume with 28-day mortality in sepsis patients:

A retrospective cohort study using eICU data

PONE-D-24-53001R1

Dear Dr. Hui-Zhen Tang,

We’re pleased to inform you that your manuscript has been judged scientifically suitable for publication and will be formally accepted for publication once it meets all outstanding technical requirements.

Kind regards,

Marwan Al-Nimer

Academic Editor

PLOS ONE

Additional Editor Comments (optional):

No coments
---

## [Editor Report · Acceptance letter]

PONE-D-24-53001R1

PLOS ONE

Dear Dr. Tang,

I'm pleased to inform you that your manuscript has been deemed suitable for publication in PLOS ONE. Congratulations! Your manuscript is now being handed over to our production team.

Kind regards,

on behalf of

Dr. Marwan Al-Nimer

Academic Editor

PLOS ONE
